# Argentinian Rose Petals as a Source of Antioxidant and Antimicrobial Compounds

**DOI:** 10.3390/foods13070977

**Published:** 2024-03-22

**Authors:** Sabrina Y. Baibuch, Laura I. Schelegueda, Evelyn Bonifazi, Gabriela Cabrera, Alicia C. Mondragón Portocarrero, Carlos M. Franco, Laura S. Malec, Carmen A. Campos

**Affiliations:** 1Departamento de Industrias, Facultad de Ciencias Exactas y Naturales, Universidad de Buenos Aires, Buenos Aires C1428EGA, Argentina; baibuchsabrina@hotmail.com (S.Y.B.); laura.schelegueda@gmail.com (L.I.S.); 2Departamento de Química Orgánica, Facultad de Ciencias Exactas y Naturales, Universidad de Buenos Aires, Buenos Aires C1428EGA, Argentina; evelynbonifazi@gmail.com (E.B.); gabyc@qo.fcen.uba.ar (G.C.); malec@qo.fcen.uba.ar (L.S.M.); 3Instituto de Tecnología de Alimentos y Procesos Químicos (ITAPROQ), Facultad de Ciencias Exactas y Naturales, Ciudad Universitaria, Consejo Nacional de Investigaciones Científicas y Técnicas, Buenos Aires C1428EGA, Argentina; 4Unidad de Microanálisis y Métodos Físicos Aplicados a la Química Orgánica (UMYMFOR), CONICET-Universidad de Buenos Aires, Buenos Aires C1428EGA, Argentina; 5Departamento de Química Analítica, Nutrición y Bromatología, Facultad de Ciencias Veterinarias, Universidad de Santiago de Compostela, 27002 Lugo, Spain; alicia.mondragon@usc.es (A.C.M.P.); carlos.franco@usc.es (C.M.F.)

**Keywords:** rose petals, antimicrobial, antioxidant, bioactive compounds, HPLC-ESI-QTOF/MS

## Abstract

The total phenolic, flavonoid, and anthocyanin contents were evaluated in 11 cultivars of Argentinian roses of different colors. HPLC-ESI-QTOF/MS was used to identify the components where ellagic and quinic acids, quercetin, and kaempferol glycosylated derivatives were found. The phenolic contents ranged from 78.8 ± 3.2 to 203.4 ± 3.1 mg GAE/g dw, the flavonoid content ranged from 19.1 ± 3.8 to 125.9 ± 6.5 mg QE/g dw, and the anthocyanin content ranged from less than 0.01 to 5.8 ± 0.1 mg CE/g dw. The dark red cultivars exhibited the greatest levels of the analyzed compounds and of the antioxidant activities, even higher than those of certain plants known for their high phenolic contents and antioxidant activity. Moreover, the addition of these extracts decreased the population of *L. innocua* and *P. aeruginosa* to undetectable levels 24 h after inoculation. Rose petal extracts, mainly those with a dark red color, can be used as natural additives in food, feed, and cosmetics, as they contain a high proportion of bioactive compounds with antioxidant and antimicrobial effects.

## 1. Introduction

Flowers, particularly roses (*Rosa* spp.), contain high concentrations of phenolic compounds [1,2], which are important secondary metabolites of plants, involved in their growth and reproduction, as well as in defensive processes against pathogens, predators, or ultraviolet radiation. These compounds can be classified into multiple subclasses, such as flavonoids, phenolic acids, stilbenes, and lignans [3]. These compounds exhibit mainly antioxidant and antimicrobial activities [4]. The antioxidant capacity of these materials is defined by their ability to trap free radicals, donate hydrogen atoms or electrons, and chelate metal ions [5]. Moreover, the described antimicrobial mechanisms are based on damage to the bacterial cell membrane, enzymatic modifications, disruption of DNA and RNA, and protein synthesis, among others [4].

There are reports indicating the antimicrobial and antioxidant effects of phenolics extracted from rose petals [1,6,7]. However, to the best of our knowledge, there is no available information on the phenolics contained in Argentinian rose cultivars or on their antioxidant and antimicrobial activities. Furthermore, roses are among the most consumed edible flowers [8]. Currently, rose petals are incorporated into various foods, such as yogurts, jams, teas, and cakes, as bioactive ingredients, natural colorants, and flavor agents [9,10]. 

The growing interest in natural additives highlights the potential of using phenolic compounds extracted from plants as natural antioxidants and antimicrobial agents. Understanding the phenolic composition of rose extracts is crucial for evaluating their potential use as additives in foods or cosmetics.

In Argentina, one of the main productive areas dedicated to floriculture is located in the town of San Pedro, Buenos Aires Province. There are many plant nurseries, with roses being one of the most commercialized flowers [11]. During the growth of rose plants, until they reach the required size for sale, several flowerings occur, which are discarded. However, these flowers could be used as a source of bioactive compounds contributing to a reduction in waste.

This study aimed to analyze hydroalcoholic extracts obtained from petals of 11 Argentinian rose cultivars of different colors and to evaluate the antimicrobial and antioxidant capacities of these extracts.

## 2. Materials and Methods

### 2.1. Chemicals

Gallic acid (GA) and Folin–Ciocalteu reagent were bought from Merck (Darmstadt, Germany). 6-Hydroxy-2,5,7,8-tetramethylchromane-2-carboxylic acid (Trolox), 2,4,6-tri (2-pyridyl)-s-triazine (TPTZ), and 2,2-diphenyl-1-picryl-hydrazyl (DPPH) were obtained from Sigma (Steinhelm, Germany). The cyanidin-3-O-glucoside (cyd-3-glu) standard was purchased from Fluka (St. Louis, MO, USA). The absolute ethanol was obtained from Biopack (Buenos Aires, Argentina). All the other chemicals utilized were of analytical quality. All culture media were purchased from Biokar (Biokar Diagnostics, Beauvais, France).

### 2.2. Plant Material

Petal flowers from the *Rosa* spp. genus were manually harvested in December 2021 from San Pedro nurseries. A total of 11 pesticide-free rose cultivars were analyzed: *Lovely Red*, *Papa Meilland*, *Lilli Marleen*, and *Oklahoma*, which have dark red coloration; *Les Amour*, *Caprice*, *Farandole*, and *Montezuma*, which have pink tones; and *Abbage*, *Prestigge Delio*, and *Iceberg*, which have light colors (Figure 1). Drying was carried out in an Edgware-Middlesex M008H104 oven (Edgware, London, UK) at 65 °C air flow for 1.5 h. The dehydrated petals were ground, sieved through a No. 20 A.S.T.M. mesh (0.850 mm), and stored in amber-colored jars at −18 °C until analysis. The final moisture level was measured using the A.O.A.C. 920.151 (1990) method through vacuum drying in an oven at 65 °C until a constant weight was achieved.

### 2.3. Extraction Procedure

The extraction of ground dehydrated petals was carried out using hydroethanolic solvent (38% ethanol) in a water bath for 30 min at 75 °C with ultrasound assistance (275 W, 50/60 Hz). The solid–solvent ratio was 1:40 for the microbiological analysis, while for the other determinations it ranged from 1:250 to 1:500 g/mL The extraction conditions were previously optimized using response surface methodology [12]. After the extraction, the mixture was centrifuged at 6440× *g* for 15 min at 4 °C, after which the supernatant was separated from the solid residue. The pellet was extracted again under identical conditions. The two supernatants were combined, and solvent was added to reach the final volume. All extractions were performed in duplicate and were used to evaluate the antioxidant and antimicrobial activity; total phenolic, flavonoid, and anthocyanin contents; and to identify each phenolic compound by HPLC-ESI-QTOF/MS analysis. These analyses were carried out in triplicate.

### 2.4. Total Phenolic Content

The detailed procedure outlined by Baibuch et al. [12], based on the Folin–Ciocalteu method [13], was followed. The extract was mixed with the Folin–Ciocalteau reagent. Subsequently, saturated sodium carbonate solution was added, and after 2 h of incubation at 30 °C, it was measured spectrophotometrically at 750 nm (UV/Vis Lamda 25; Pekin Elmer, Waltham, MA, USA). The results are expressed as mg GAE/g dw.

### 2.5. Antioxidant Activity

The antioxidant activity was determined using two methods: a reduction in the DPPH radical and determination of the ferric reducing antioxidant power (FRAP). The results were quantified as mg Trolox equivalents (TE)/g dw.

The DPPH assay was assessed according to Baibuch et al. [12], using the method reported by Brand-Williams et al. [14], which involves measuring the mixture of DPPH reagent and extract at 517 nm after 30 min at 30 °C.

The antioxidant activity was determined using the FRAP method as described by Benzie and Strain [15] with slight changes. An aliquot of 200 μL of the diluted extract was mixed with 1.8 mL of FRAP reagent. After 120 min, the Perkin Elmer spectrophotometer was used to measure the absorbance at 593 nm. 

### 2.6. Anthocyanin Content

The anthocyanin content was determined using the differential pH (solutions at pH 1.0 and pH 4.5 at 510 nm and 710 nm) according to the methodology described by Baibuch et al. [12]. The results are expressed as mg CE/g dw.

### 2.7. Flavonoid Content

Flavonoid quantification was performed using the colorimetric method described by Loizzo et al. [16]. An aliquote (0.5 mL) of the extract was mixed with 2.2 mL of distilled water and 0.15 mL of 5% sodium nitrite. After 5 min, 0.15 mL of AlCl_3_ (10% *w*/*v*) were added and 6 min later, 1 mL of 1 M NaOH and 1 mL of distilled water were incorporated. The absorbance was measured at 510 nm and the flavonoid content was expressed in mg of quercetin equivalents (QE)/g dw.

### 2.8. HPLC-ESI-QTOF/MS Analysis

The phenolic compounds in all the extracts were analyzed using HPLC-ESI-QTOF/MS. The analysis was carried out using an Agilent 1200 HPLC instrument coupled to a Bruker QTOF-QII high-resolution mass spectrometer with an electrospray ionization source. The sample was separated with a C18(2) Phenomenex Luna^®^ 3 µm 100 Å, 100 × 2 mm column using a mobile phase consisting of 0.1% *v*/*v* formic acid in water (solution A) and methanol (solution B), delivered at a flow rate of 0.3 mL/min. The injection volume was 5 µL. The following elution program was carried out: 0 min, 90% A, 10% B; 2 min, 90% A, 10% B; 25 min, 25% A, 75% B; 26 min, 0% A, 100% B; 44 min, 0% A, 100% B; 45 min, 90% A, 10% B; and 50 min, 90% A, 10% B. Nitrogen was used as the nebulizer gas, and the mass scan in negative- and positive-ion modes ranged from 100 to 1000 *m*/*z*. The peak areas of each compound in the chromatograms were compared among the different cultivars, and the differences and/or similarities between them were determined.

### 2.9. Antimicrobial Activity In Vitro

#### 2.9.1. Indicator Strains and Culture Conditions

To evaluate the antimicrobial activity of the extracts, the following strains were used: *Listeria innocua* ATCC 33090, a surrogate for the pathogen *Listeria monocytogenes*, and *Pseudomonas aeruginosa* ATCC 9027, which are commonly Gram-negative deteriorative bacteria. Both were stored at −80 °C in nutrient broth supplemented with 20 g/100 g of glycerol. Before use, each bacterium was incubated in trypticase soy broth (TSB) twice for 24 h at 30 °C. To achieve the desired concentration, turbidity was determined by the McFarland method, and necessary dilutions were then made with peptone water.

#### 2.9.2. System Preparation

The test involved 1.5 mL of extract (after ethanol evaporation) and 3.5 mL of TSB. The systems were inoculated with 10^4^ CFU/mL of each microorganism. All the samples were kept at 30 °C for 24 h under stirring. Control samples with identical compositions but without inoculation were also prepared to assess the microbiological status of the extract and the handling conditions. Additionally, inoculated samples were prepared by substituting the extract (1.5 mL) with sterile distilled water in a final volume of 5 mL to ensure uniform access to nutrients in the culture medium. All the systems were stored in duplicate.

#### 2.9.3. Antimicrobial Activity

Antimicrobial activity was evaluated by the plate count method after 24 h of system incubation at 30 °C. Aliquots of each system were serially diluted with peptone water and poured onto bacterial plate counting agar plates. The plates were incubated at 30 °C for 48–72 h, after which the CFUs were counted. The measurements were conducted in duplicate.

### 2.10. Data Analysis

The experimental data were evaluated by a one-way analysis of variance (ANOVA) with InfoStat Software 2020 version, (University of Córdoba, Córdoba, Argentina). To perform multiple comparisons, the Tukey test was used. A probability of error less than 5% was considered to indicate statistical significance (*p* value < 0.05). To establish the correlation between the various parameters, linear regression was used, and the correlation coefficient (R^2^) was calculated.

## 3. Results and Discussion

### 3.1. Bioactive Compounds

As shown in Figure 2, the total phenolic, flavonoid, and anthocyanin contents were evaluated in 11 rose cultivars. The phenolic content ranged from 78.8 ± 3.2 to 203.4 ± 3.1 mg GAE/g dw. The values for pink and light petals obtained in this study are in line with the findings of Alizadeh and Fattahi [17] and Chen et al. [18] for pink and purple cultivars of rose flowers. In their studies, the total phenolic content ranged from 65 to 165 mg GA/g dw. On the other hand, the values for the dark red cultivars analyzed in this study were higher. Furthermore, these values were greater than those established by Chen et al. [18] for *Rosmarinus officinalis* L. (47.2 ± 0.6 GAE/g dw), widely recognized as one of the species with the highest phenolic compound contents.

The flavonoid content varied from 19.1 ± 3.8 to 125.9 ± 6.5 mg QE/g dw. It must be stressed that these values were higher than those reported by Alizadeh and Fattahi [17].

The anthocyanin content varied from less than 0.01 to 5.8 ± 0.1 mg CE/g dw. These findings are consistent with previous studies on rose petals, in which the anthocyanin content was within the range of 0.05 to 6 mg CE/g dw [2,17,19]. The low values of light-colored Argentinian roses were also in agreement with those of Yeon and Kim [20] and dos Santos [2], who reported the absence of anthocyanin or a content of 0.05 mg CE/g dw, respectively, for white rose cultivars. Our research revealed that anthocyanins constitute approximately 1% of the total phenolic compounds in dark red cultivars. Despite this low percentage, the impact of anthocyanin content on petal color was notable since the dark red cultivars presented the highest concentration of anthocyanins.

Among the dark red cultivars, *Lovely Red*, *Papa Meilland*, *Lilli Marleen*, and *Oklahoma* exhibited the highest values for all the bioactive compounds analyzed, while the lowest values were associated with light-colored petals.

### 3.2. Identification of Phenolic Compounds

Accurate mass measurements, analysis of mass fragmentation patterns in negative and positive ionization mode, and a comprehensive review of the literature previously mentioned in Baibuch et al. [12] were carried out. The findings are presented in Table 1, which provides information on the tentatively identified compounds, including their retention times, molecular formulas, MS/MS fragmentation and mass‒charge ratios (*m*/*z*). As an example, Figure 3 shows the chromatogram of the *Les Amour* cultivar, where the peaks of the identified compounds can be observed.

The results showed that ellagic acid was present in all the cultivars analyzed. Ellagic acid, a natural flavonoid and dimeric derivative of gallic acid, was found to be a significant constituent of various medicinal plants, fruits, vegetables and particularly *Rosa rugosa* [21]. It has been demonstrated to have bioactive properties, such as antioxidant and antimicrobial activities. In plants, it was primarily found esterified with sugars as part of hydrolyzable tannins known as ellagitannins [22]. In some of the cultivars analyzed, an unidentified ellagitannin was found.

Additionally, some flavonols, particularly those derivatives of quercetin and kaempferol, were identified in their glycosylated forms. Several studies have also informed that quercetin, kaempferol, and their derivatives are commonly present in flowers [23], especially in diverse cultivars of roses [24,25]. These compounds have at least 270 different forms of glycosylation [26].

In addition, quinic acid, which is not a phenolic compound but a cyclohexane carboxylic acid, was found in all the analyzed cultivars.

The dark red cultivars were also analyzed in the positive mode on account of their high total anthocyanin contents. Two anthocyanins were identified: cyanidin-3,5-O-diglucoside and malvidin-3-acetylglucoside-4-vinylphenol. Cyanidin and malvidin derivatives are among the most common anthocyanidins found in plants [26]. Cyanidin-3,5-O-diglucoside has also been reported in previous studies as being present in rose petals [19,23]. 

While very few compounds were present in all the cultivars, most were detected in only some of them. Therefore, each cultivar exhibited a distinct composition profile without revealing a pattern among petals of similar colors. Furthermore, the comparison of the peak areas among the different analyzed cultivars revealed significant variations in the concentrations of the identified compounds. Among the rose cultivars, dark red had the highest concentration, followed by pink, with white flowers having the lowest concentration, as was also noted for total phenolic, flavonoid and anthocyanin contents.

### 3.3. Antioxidant Activity

Figure 4 shows the antioxidant capacity of the floral extracts obtained from the 11 evaluated cultivars by applying the DPPH radical scavenging activity and FRAP methods. The dark red cultivars showed the highest antioxidant activities, ranging from 506.8 ± 17.6 to 663.0 ± 3.2 mg TE/g dw, as measured by the DPPH assay, and from 699.8 ± 21.6 to 789.3 ± 16.9 mg TE/g dw by the FRAP method in *Oklahoma* and *Lovely Red*, respectively. Previous studies have shown the high antioxidant capacity of roses, which rank them as having the highest antioxidant capacity among many other flowers [1,24]. The antioxidant activity found in the analyzed Argentinian cultivars was greater than that reported for roses by Ginova et al. [19] and VanderJagt et al. [27]. Notably, the antioxidant effect of these extracts was even greater than that of *Rosmarinus officinalis* and *Illex paraguariensis* [27], which are recognized for their antioxidant activity.

The results obtained with both methodologies showed a high correlation (R^2^ = 0.9731). In addition, the coefficient of determination between the antioxidant capacity and the phenolic content (Section 3.1) was also high for both methods (R^2^ = 0.9857 with DPPH and R^2^ = 0.9579 with FRAP). Other studies on edible flowers have also shown a good correlation between phenolic content and antioxidant capacity [1,10,28], revealing the strong influence of these compounds on antioxidant activity. There was also a good relationship between antioxidant activity and the concentration of flavonoids (R^2^ = 0.6631 with DPPH and R^2^ = 0.7359 with FRAP) and particularly with the concentration of anthocyanins (R^2^ = 0.7455 with DPPH and R^2^ = 0.7829 with FRAP). The lower correlation coefficients of flavonoids and anthocyanins compared to those of total phenolics may be due to the antioxidant contribution of other phenolics different from flavonoids, such as ellagic acid, which was identified in all the cultivars (Section 3.2). In particular, it was reported that ellagic acid exhibited a high in vitro antioxidant activity [29], even higher than ascorbic acid [22]. According to Zheng et al. [23], anthocyanins and phenolic acids are the primary contributors to antioxidant activity in China roses. Therefore, the higher concentration of anthocyanins in the dark red cultivars (Section 3.1) may also contribute to their strong antioxidant capacity, despite their low proportion of total phenolics. Ferreira et al. [30] studied the antioxidant properties of eight flavonoids, revealing that kaempferol and quercetin exhibit strong and comparable antioxidant capacity attributed to their similar structures. The specific structure of phenolic compounds plays a more significant role in the antioxidant effects than quantitative aspects [31,32]. On the other hand, quinic acid did not exhibit antioxidant activity, as evaluated by multiple in vitro methods [33]. This could be attributed to the absence of an aromatic structure, which is necessary for the stabilization of the unpaired electron by resonance.

The dark red cultivars exhibited greater antioxidant activity because of the higher concentration of bioactive compounds. However, the complexity of the antioxidant mechanisms prevents the attribution of the antioxidant effect to any specific compound but rather to the combination of the different compounds.

### 3.4. Antimicrobial Activity

Figure 5 shows the effect of adding extracts from the different cultivars on the growth of *L. innocua* (panel A) and *P. aeruginosa* (panel B). In the systems free of extract (the control), the population of both bacteria increased six log cycles, reaching more than 6. 10^10^ CFU/mL after 24 h of storage. In the case of *L. innocua*, the addition of extracts of the dark red cultivars *Lilli Marleen*, *Oklahoma*, *Lovely Red*, and *Papa Meilland* to broth promoted a decrease in the population to undetectable levels (less than 10 CFU/mL) after 24 h of storage. The addition of the *Farandole* extract resulted in an increase of one log cycle, while the addition of the *Iceberg*, *Prestigge Delio*, and *Les Amour* extracts led to a two-log cycle increase. The extracts obtained from the *Montezuma*, *Caprice*, and *Abbage* cultivars exhibited the lowest inhibitory activity, since the population reached counts in the range of 2.3 × 10^7^ to 6.2 × 10^8^ CFU/mL (Figure 5A).

Regarding the effect on *P. aeruginosa* (Figure 5B), the addition of the previously mentioned extracts of dark red cultivars reduced the inoculated population to less than 10 CFU/mL after 24 h of storage, while the *Iceberg*, *Prestigge Delio*, and *Abbage* extracts did not modify the population level. The control system reached a population count higher than 6 × 10^10^ CFU/mL after 24 h, increasing by six log cycles. The *Les Amour* extract led to a one-log cycle increase in the inoculated population at 24 h. The *Farandole*, *Montezuma*, and *Caprice* extracts were able to reduce the multiplication of *P. aeruginosa* compared to that of the control, but the population counts at 24 h were still high, revealing the limited inhibitory action of these extracts.

In non-inoculated systems containing the extracts, no microbial growth was observed, confirming the absence of contaminating microbiota under the working conditions and the lack of indigenous biota in the extract.

Notably, the dark red cultivars *Lovely Red*, *Oklahoma*, *Papa Meilland*, and *Lilli Marleen*, in addition to having the highest phenolic contents and antioxidant activities, also exhibited the highest antimicrobial activity against *L. innocua* and *P. aeruginosa*. When comparing the results obtained for both bacteria evaluated, it was observed that extracts of *Iceberg*, *Prestigge Delio*, *Abbage*, *Les Amour*, and *Montezuma* inhibited *P. aeruginosa* more strongly than *L. innocua*. *Farandole* exhibited the opposite trend. However, for the other cultivars, the inhibitory response was the same for both microorganisms. According to Bouarab-Chibane et al. [4], *P. aeruginosa* is one of the most resistant bacteria to the action of polyphenols, making the results obtained with the evaluated rose extracts highly relevant. Previous studies have reported the antimicrobial effect of rose extracts on Gram-positive and Gram-negative bacteria, including *L. monocytogenes* and *P. aeruginosa* [34,35,36].

The antimicrobial properties of phenolic compounds have previously been reported and are related to their molecular structure [37]. Specifically, Manso et al. [38] provided detailed information on this topic, highlighting that flavonoids and hydrolysable tannins (e.g., gallotannins and ellagitannins) are the most active polyphenols. Research has shown that phenolic acids can disrupt membrane integrity, leading to the release of essential intracellular components. Furthermore, the effectiveness of these agents is also influenced by the type of microorganism and the strain. Certain phenolics inhibit the growth of Gram-negative bacteria by damaging the outer membrane [39,40], while others do not have inhibitory effects [37]. Ellagic acid, which was identified in all studied cultivars, has been shown to have inhibitory effects on pathogenic fungi, Gram-positive bacteria, and Gram-negative bacteria [41]. It was found to have antimicrobial action against *P. aeruginosa* and *L. monocytogenes* in particular [19]. Flavonoids are known to form complexes with soluble proteins located outside or inside the cell walls of bacteria. In particular, quercetin, one of the most studied flavonoids, has been found to exhibit antibacterial activity [4]. In line with these observations, it should be noted that four quercetin derivatives were detected in most of the Argentinian cultivars analyzed.

Quinic acid, which was also identified in all the evaluated cultivars, was mentioned to exhibit antimicrobial activity, despite not having a phenolic structure. In particular, Ercan et al. [33] reported that this compound was highly effective against Gram-negative bacteria (*K. pneumoniae*, *P. aeruginosa*, *E. coli*) and Gram-positive bacteria (*S. pyogenes*, *S. aureus*). The authors stated that the effectiveness of quinic acid was due to its ability to decrease membrane flexibility and inhibit cell wall formation. Furthermore, Lu et al. [41] reported that quinic acid can inhibit the formation of bacterial biofilms by *P. aeruginosa* and may be effective against *P. aeruginosa*-related infections.

Antimicrobial and antioxidant activity were greater for the dark red cultivars than for the other cultivars, and this trend could be linked to the higher concentration of bioactive compounds in the former. Furthermore, antimicrobial activity could not be attributed to a specific compound since numerous compounds identified in the extracts exhibited antimicrobial action. It is worth noting that a larger amount of rose powder was necessary for the microbiological tests than for the antioxidant activity assays; therefore, the antioxidant action of the extracts proved to be more effective than the antimicrobial one.

## 4. Conclusions

Argentinian rose petal extracts are rich in bioactive compounds, which can lead to antioxidant and antimicrobial effects. These positive findings support the use of floral production waste, helping to revalue it as a byproduct. The dark red roses have higher total phenolic contents and even greater antioxidant activity than other well-known antioxidant plant sources. They also exhibit high antimicrobial activity. The large amount of phenolic compounds with antioxidant and antimicrobial activities found in roses and the complexity of the mechanisms involved in the mentioned activities make it impossible to attribute such activities to any specific compound but rather to their combination. According to the obtained results, dark red roses are the most suitable for potential use as natural additives in food, feed, or cosmetics.

## Figures and Tables

**Figure 1 foods-13-00977-f001:**
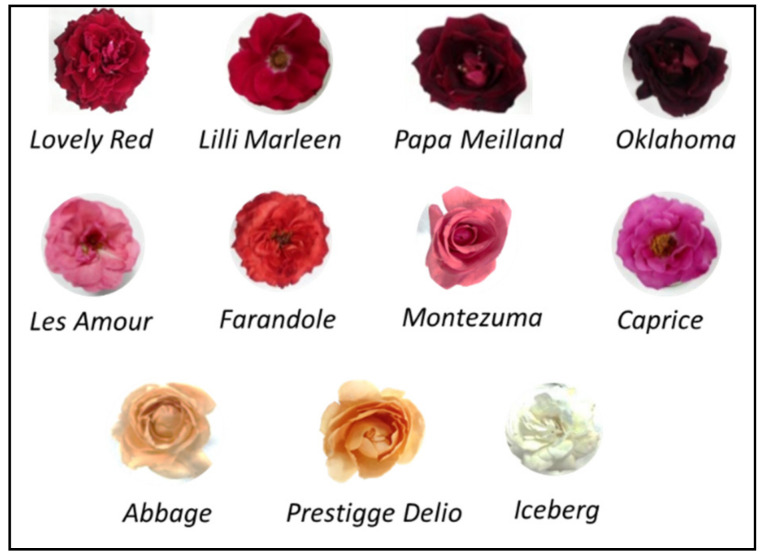
The images of the 11 Argentinian rose cultivars analyzed were grouped according to color: the top row consisted of dark red, the second row consisted of pink, and the bottom row consisted of light cultivars.

**Figure 2 foods-13-00977-f002:**
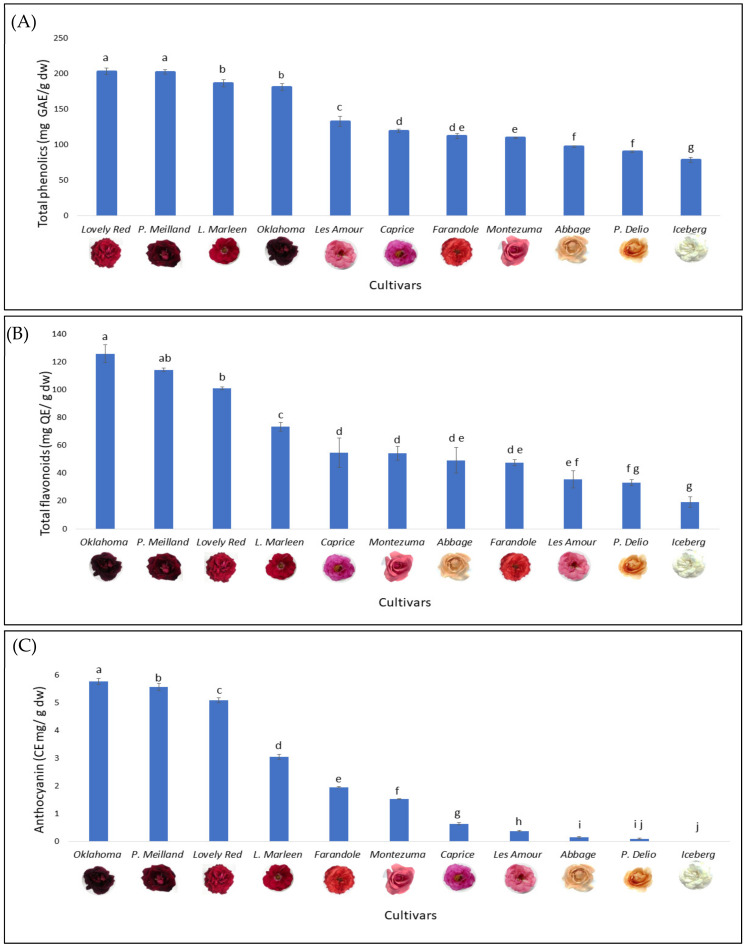
Contents of total phenolics (**A**), flavonoids (**B**) and anthocyanins (**C**) in 11 Argentinian rose cultivars. Different letters denote significant differences between cultivars.

**Figure 3 foods-13-00977-f003:**
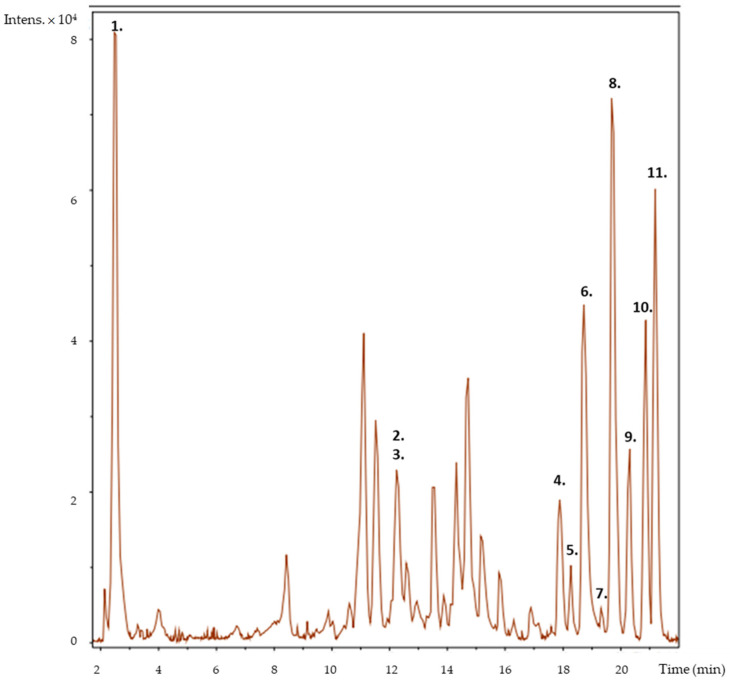
Chromatogram of the *Les Amour* cultivar showing the presence of the following compounds: 1. quinic acid; 2. galloyl-bis-HHDP-hexose; 3. unknown ellagitannin; 4. quercetin-O-pentoside; 5. quercetin-O-hexoside; 6. ellagic acid; 7. quercetin-O-rhamnoside; 8. quercetin-O-galloyl rhamnoside; 9. kaempferol-O-pentoside; 10. kaempferol-O-hexosyl-deoxyhexoside; 11. kaempferol-O-deoxyhexoside.

**Figure 4 foods-13-00977-f004:**
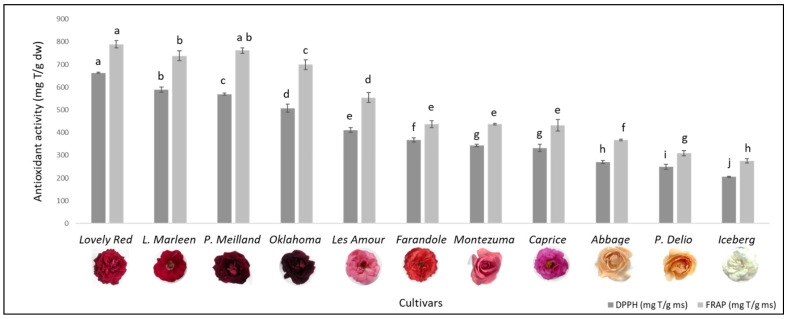
Antioxidant activity of floral extracts from 11 Argentinian rose cultivars measured by the reduction in the DPPH radical (dark columns) and FRAP method (light columns). Different letters represent significant differences between values for the same determination but for different cultivars.

**Figure 5 foods-13-00977-f005:**
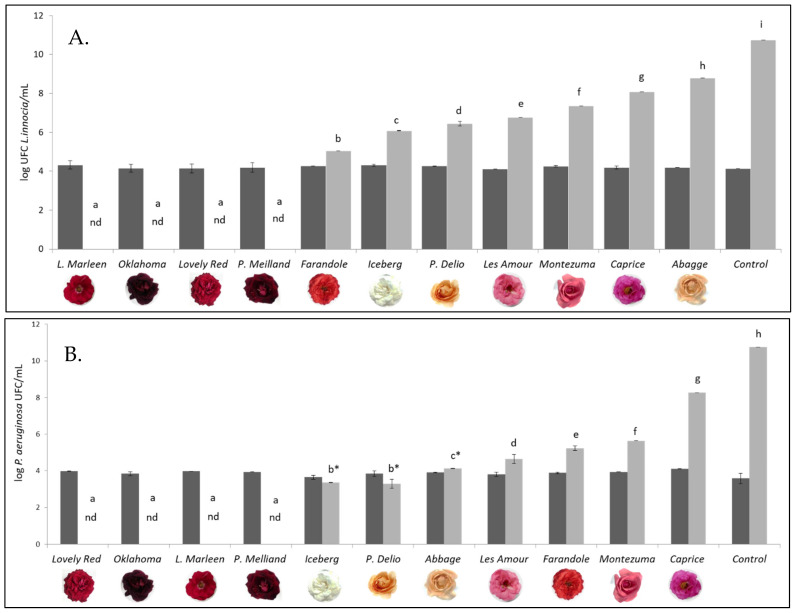
Effects of the extracts from 11 Argentinian rose cultivars on the growth of *L. innocua* (**A**) and *P. aeruginosa* (**B**). Initial count of *L. innocua* (dark bars); 24 h count (light bars). Different letters denote significant differences between cultivars. nd: not detectable. * indicates the absence of significant differences between 0 and 24 h for the same cultivar.

**Table 1 foods-13-00977-t001:** Tentative identification of compounds in floral extracts from 11 Argentinian rose cultivars of different colors by HPLC-ESI-QTOF/MS.

Negative Mode						
N◦	Rt. (min)	Formula	Compound	[M-H]− (*m*/*z*)	MS/MS	Flower Cultivar
1	2.5	C_7_H_12_O_6_	Quinic acid	191.0558	171, 155, 137, 127	All cultivars
2	12.2	C_41_H_28_O_26_	Galloyl-bis-HHDP-hexose	935.0772	451	Red flowers: *Lilli Marleen*, *Papa Meilland, Oklahoma* Pink flowers: *Les Amour*, *Farandole* White flowers: *Iceberg*
3	12.5	C_41_H_26_O_26_	Unknown ellagitannin	466.0291 *	301, 451	Red flowers: *Lilli Marleen*, *Papa Meilland* Pink flowers: *Les Amour*, *Montezuma*, *Caprice* White flowers: *Iceberg*
4	17.8	C_20_H_18_O_11_	Quercetin-O-pentoside	433.0762	410, 300	Red flowers: *Papa Meilland* Pink flowers: *Les Amour*, *Caprice*, *Montezuma* White flowers: *Abagge*, *Prestigge Delio*, *Iceberg*
5	18.1	C_21_H_20_O_12_	Quercetin-O-hexoside	463.0899	410, 300	Pink flowers: *Les Amour* White flowers: *Iceberg*
6	18.6	C_14_H_6_O_8_	Ellagic acid	300.9995	229, 210, 201, 185, 173	All cultivars
7	19.5	C_21_H_20_O_11_	Quercetin-O-rhamnoside	447.0930	410, 300, 271, 178, 151	All cultivars
8	19.7	C_28_H_24_O_15_	Quercetin-O-galloyl rhamnoside	599.1042	447, 410, 313, 285, 226, 169	Pink flowers: *Les Amour*, *Montezuma*, *Caprice* White flowers: *Prestigge Delio*, *Iceberg*
9	20.2	C_20_H_18_O_10_	Kaempferol-O-pentoside	417.0814	284, 255, 227	Red flowers: *Oklahoma* Pink flowers: *Montezuma*, *Caprice*, *Farandole*, *Les Amour*
10	20.7	C_27_H_30_O_15_	Kaempferol-O-hexosyl-deoxyhexoside	593.1507	410, 285	Pink flowers: *Les Amour*, *Montezuma*, *Farandole* White flowers: *Abagge*, *Prestigge Delio*, *Iceberg*
11	21.1	C_21_H_20_O_10_	Kaempferol-O-deoxyhexoside	431.1003	410, 284, 255, 227	Red flowers: *Lilli Marleen*, *Lovely Red*, *Papa Meilland*, *Oklahoma* Pink flowers: *Les Amour*, *Montezuma*, *Farandole*, White flowers: *Abagge*, *Prestigge Delio*, *Iceberg*
Positive mode						
N◦	Rt. (min)	Formula	Compound	[M-H] + (*m*/*z*)	MS/MS	Flower cultivar
1	8.6	C_27_H_31_O_16_	Cyanidin-3,5-O-diglucoside	611.1609	287, 449	All dark red cultivars
2	10.0	C_33_H_31_O_14_	Mavidin-3-acetylglucoside-4-vinylphenol	651.1711	489, 471, 379, 327, 309, 277, 185	All dark red cultivars

* Corresponds to the ion [M-2H]2−, HHDP: hexahydroxydiphenoyl.

## Data Availability

The original contributions presented in the study are included in the article, further inquiries can be directed to the corresponding author.

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
