# Peer review of "Argentinian Rose Petals as a Source of Antioxidant and Antimicrobial Compounds"

_foods, 2024, doi:10.3390/foods13070977_

Round 1

Reviewer 1 Report

Comments and Suggestions for Authors

The main object of this research is the plant species from genus Rosa spp. The authors selected 11 Argentinian rose cultivars of different colors and evaluated the antioxidant and antimicrobial potential of extracts made from the flower petals. They applied standard procedures to obtain the aqua-ethanolic extracts, and standard methods to determine in vitro antioxidant and antimicrobial activities. The authors applied a modern analytical method and technique like HPLC-ESI-QTOF/MS for identification of the extracted phenolic compounds.

Rosa plant species have long been known for their ingredients with cosmetic and antioxidant properties, and applied in the fragrance production and medicine (many researcher teams worked on this scientific field as cited by the authors). Nerveless, the study subject is actual, because of the research object picked up – Argentina roses are traditionally produced, and their properties and qualities have not been thoroughly studied. It is relevant and interesting, because of its importance for practice. (The phytochemistry is the future of the pharmacy.) In addition, rose species are edible flowers and this research will promote this plant as an edible and healthy decoration.

The paper is well written, the text is clear and easy to read. The authors have done a great deal of work, specifying the phenolic profile of the extracts from roses, widely cultivated in Argntina and also very interesting information of their effect of in vitro antimicrobial activity.

I have some suggestion.

General:

·       Please, introduce all abbreviation used in the manuscript. There are many abbreviations in the main text, so for better text understanding und reading, I suggest the authors to add an abbreviation interpretation section.

·       I did not found any correlation evaluated between the identified compounds with proven antioxidant properties and the antioxidant and antimicrobial potential of the extract obtained. Please, add such information. This is more useful.

·       Please, add some chromatogram from the HPLC measurement in the Results section.

·       Please, check all cited references source in the whole manuscript. E.g. Line 216: “(16, 20)”, must be [16, 20].

·       Please, check the Reference section. There are references introduced not acc. Foods guidelines.

1.     1. Introduction

Lines 48-50: ”Understanding the phenolic composition of rose extracts is crucial for evaluating their potential use as additives in foods or cosmetics.”

Please, add more info about the application of this Rosa species in the food proceeding and cite references. Moreover, the main scope of the journal “Foods” are the food proceeding, quality, analysis etc.

2.     2.5. Total phenolic content:

There is cited method applied. Please, add some info about procedure details.

Line 99: “mg GA/g– is it from gallic acid equivalents? Please, introduce this abbreviation. It is correct mg GAE/g. Correct it in the whole manuscript.

3.     2.6. Antioxidant activity: Please, add some info about procedure details of DPPH methods, and introduce the dimension used. Correct is mg TE/g (TE is from trollox equivalents), and correct it in the whole manuscript.

4.     2.7. Anthocyanin content:

Please, add some info about procedure details of the method.

Line: 112: ”cyd 3-glu”: please insert the whole name of the compound, cyanidin 3-glucoside. In the dimension of the values obtained can be note as mgCE/g (CE from cyanidin equivalents)

5.     2.8. Flavonoid content:

Please, add some info about procedure details of the method.

Line 115: “mg of quercetin per g dry Please, introduce the dimension used finely. Correct is mg QE/g (QE is from quercetin equivalents). Please, correct it in the whole manuscript.

6.     2.9. HPLC-ESI-QTOF/MS analysis, Line: 121: “Phenomenex Luna® 3 μm C18(2) 100 Å 121 column..”

Please, add all parameters of the column, e.g. length and i.d.

7.     2.11. Data analysis. Line 155: “The experimental data were analyzed by analysis of variance (ANOVA) with InfoStat Software (University of Córdoba, Argentina).”

Please, classify the ANOVA test. Which one? One-way, two-way, MANOVA, factorial ANOVA?

8.     4. Conclusions”: Please, insert more summarized info, e.g. which compounds with proven antioxidant properties are identified. Is there some correlation between determined compounds and antioxidant and antimicrobial potential etc.

Author Response

Thanks for your letter concerning the revision of our manuscript. The criticisms have been very helpful to improve the manuscript’s quality. In this letter you will find a detailed list of changes introduced in the revised manuscript according to your suggestions.

Reviewer suggestions

General:

  • Please, introduce all abbreviation used in the manuscript. There are many abbreviations in the main text, so for better text understanding und reading, I suggest the authors to add an abbreviation interpretation section.

We added an abbreviation section placed after the keywords.

  • I did not found any correlation evaluated between the identified compounds with proven antioxidant properties and the antioxidant and antimicrobial potential of the extract obtained. Please, add such information. This is more useful.

The objective of this study was to analyze the composition of  hydroalcoholic extracts obtained from petals of 11 Argentinian rose cultivars of different colors and to evaluate the antimicrobial and antioxidant capacities of these extracts. The final purpose was to gain knowledge about possible uses of these extracts. As a result of the work, we identified in the extracts several compounds which according to previous studies were proven to exert antioxidant and/or antimicrobial activity. We did not evaluate each pure compound alone since we were interest in the activity of the extracts containing the mixture of them. Furthermore, it is well known that in a mixture antioxidant and antimicrobial activity is the result of the interaction among the different compounds present more than the specific activity of a compound.

Please, add some chromatogram from the HPLC measurement in the Results section.

We added as an example a chromatogram  of the Les Amour cultivar, where the peaks of the identified compounds can be observed (see Figure 3).

  • Please, check all cited references source in the whole manuscript. E.g. Line 216: “(16, 20)”, must be [16, 20].

We changed the round brackets by the square brackets (see line 246).  

  • Please, check the Reference section. There are references introduced not acc. Foods guidelines.

 We revised all the section and made some changes in references numbers 8, 23 and 31.

  1. Introduction

Lines 48-50: ”Understanding the phenolic composition of rose extracts is crucial for evaluating their potential use as additives in foods or cosmetics.”

Please, add more info about the application of this Rosa species in the food proceeding and cite references. Moreover, the main scope of the journal “Foods” are the food proceeding, quality, analysis etc.

We added a paragraph about food uses of rose petals (lines 47-49).

2.5. Total phenolic content:

There is cited method applied. Please, add some info about procedure details.

We added additional info (see lines 107-110).

Line 99: “mg GA/g” – is it from gallic acid equivalents? Please, introduce this abbreviation. It is correct mg GAE/g. Correct it in the whole manuscript.

We made the correction.

   2.6. Antioxidant activityPlease, add some info about procedure details of DPPH methods, and introduce the dimension used. Correct is mg TE/g (TE is from trollox equivalents), and correct it in the whole manuscript.

We made the correction.

  2.7. Anthocyanin content:

Please, add some info about procedure details of the method.

We added additional info (see lines 125-126).

Line: 112: ”cyd 3-glu”: please insert the whole name of the compound, cyanidin 3-glucoside. In the dimension of the values obtained can be note as mgCE/g (CE from cyanidin equivalents)

We inserted the whole name in Materials and Methods as it was suggested (see line 68).

We also changed the expression of the corresponding results.

2.8. Flavonoid content:

Please, add some info about procedure details of the method.

We added additional info (see lines 131-134).

Line 115: “mg of quercetin per g dry Please, introduce the dimension used finely. Correct is mg QE/g (QE is from quercetin equivalents). Please, correct it in the whole manuscript.

We made the correction.

2.9. HPLC-ESI-QTOF/MS analysis, Line: 121: “Phenomenex Luna® 3 μm C18(2) 100 Å 121 column..”

Please, add all parameters of the column, e.g. length and i.d.

We added column dimensions (see lines 140-141).

  2.11. Data analysis. Line 155: “The experimental data were analyzed by analysis of variance (ANOVA) with InfoStat Software (University of Córdoba, Argentina).”

Please, classify the ANOVA test. Which one? One-way, two-way, MANOVA, factorial ANOVA?

We applied one-way ANOVA and this was added at line 176.

     4. Conclusions”: Please, insert more summarized info, e.g. which compounds with proven antioxidant properties are identified. Is there some correlation between determined compounds and antioxidant and antimicrobial potential etc.

We added a paragraph mentioning that due to the complex mechanisms of antioxidant and antimicrobial activities and the complex mixture of compounds identify in the extract which can exert activity it is not possibly to attribute the activity to any specific one but rather to the combination of the many compounds which can act synergistically (see lines 387-391).

In addition to all changes previously mentioned, we  made some changes to avoid plagiarism.

Reviewer 2 Report

Comments and Suggestions for Authors

The ms looks very interesting as the rose petals could serve as a raw material for many applications. The ms is well-structured and fairly proportional and the topics present the relevant experimental outcomes.  The chemical characterization of the petal extracts shows interesting differences, but it is not indicated if the same quantity of dehydrated petals was used for the preparation of all the extracts. If this were the case, then the study would gain a comparative feature that would further increase the importance of the ms. I would kindly ask the authors to clarify this issue as it would be also important to compare the antimicrobial activity.  Regarding the antimicrobial activity, it is again not clearly stated if the authors were focusing on the bacteriostatic and/or bactericidal features. Distinguishing between these two would be important especially if these extracts are meant for human type of applications.  

Author Response

Thanks for your letter concerning the revision of our manuscript. The criticisms have been very helpful to improve the manuscript’s quality. In this letter you will find a detailed list of changes introduced in the revised manuscript according to your suggestions.

Comments and Suggestions for Authors

The ms looks very interesting as the rose petals could serve as a raw material for many applications. The ms is well-structured and fairly proportional and the topics present the relevant experimental outcomes. 

The chemical characterization of the petal extracts shows interesting differences, but it is not indicated if the same quantity of dehydrated petals was used for the preparation of all the extracts. If this were the case, then the study would gain a comparative feature that would further increase the importance of the ms. I would kindly ask the authors to clarify this issue as it would be also important to compare the antimicrobial activity. 

We added the ratio range of powder/solvent used for chemical and microbiological analysis at lines 91-92.

Regarding the antimicrobial activity, it is again not clearly stated if the authors were focusing on the bacteriostatic and/or bactericidal features. Distinguishing between these two would be important especially if these extracts are meant for human type of applications. 

Microbiological activity was evaluated by the plate count method; therefore, the extracts exerted a bactericidal effect. Even though, they were obtained from pesticide-free rose cultivars, a toxicological evaluation is a must prior to their application in food products.

In addition to all changes previously mentioned, we  made some changes to avoid plagiarism.

Round 2

Reviewer 1 Report

Comments and Suggestions for Authors

Dear Authors,

you have complied with my suggestion. I have no comments more.

Good luck!